# Association between Legume Consumption and Risk of Hypertension in the European Prospective Investigation into Cancer and Nutrition (EPIC)-Norfolk Cohort

**DOI:** 10.3390/nu14163363

**Published:** 2022-08-16

**Authors:** Michael Hartley, Claire L. Fyfe, Nicholas J. Wareham, Kay-Tee Khaw, Alexandra M. Johnstone, Phyo K. Myint

**Affiliations:** 1Institute of Applied Health Sciences, University of Aberdeen, Polwarth Building, Foresterhill, Aberdeen AB25 2ZD, UK; 2The Rowett Institute, University of Aberdeen, Ashgrove Road West, Aberdeen AB25 2ZD, UK; 3MRC Epidemiology Unit, Cambridge CB2 9SL, UK; 4Clinical Gerontology Unit, Department of Public Health and Primary Care, Gonville and Caius College, Cambridge CB2 1TA, UK; 5Aberdeen Cardiovascular & Diabetes Centre, University of Aberdeen, Polwarth Building, Foresterhill, Aberdeen AB25 2ZD, UK

**Keywords:** legumes, hypertension, blood pressure, plant-based, protein, cardiovascular disease

## Abstract

Hypertension is a significant and preventable cardiovascular disease risk factor. Growing evidence suggests legumes have blood-pressure (BP) lowering properties. However, there is little population-based research on legume intake and hypertension risk in Western populations. The objective was to investigate the relationship between legume intake and blood pressure by using data from the European Prospective Investigation into Cancer and Nutrition (EPIC) Norfolk cohort. Further, to identify any potential legume intake that confers benefits in relation to blood pressure. We included participants who completed both 7-day food diaries to assess legume intake and undertook a first (1993–1997) and second (1998–2000) health check from the EPIC-Norfolk prospective study. Legume consumption was categorized using percentile cut off values. We used multivariate logistic regression models to calculate the odds ratio of hypertension (defined as >140 mmHg systolic and/or >90 mmHg diastolic blood pressure) at the second health check, stratified by legume intake, adjusting for antihypertensive medication use and demographic, socioeconomic and lifestyle covariates. A total of 7522 participants were included with mean age (± SD) of 58.0 ± 8.9 years. The follow-up time was 3.7 years (range: 2.1–6.6 years). Mean legume consumption was 17.3 ± 16.3 g/day. Participants in the 97th percentile of legume intake had the lowest odds of subsequent hypertension (OR: 0.71; 95% CI: 0.52, 0.96). Legume consumption between 55–70 g/day was associated with reduced odds of hypertension (OR: 0.57; 95% CI: 0.37, 0.88); sex-specific values for men and women were 0.64 (0.38, 1.03) and 0.32 (0.12, 0.88), respectively. In this UK population, legume intake of 55–70 g/day was associated with a lower subsequent risk of hypertension. Given the low legume intake in the UK and Western countries, dietary guidance to increase intake above 55 g/day may lower the burden of hypertension and associated diseases.

## 1. Introduction

Hypertension is a major risk factor for cardiovascular and renal disease worldwide [1]. The aetiology of elevated blood pressure is linked to genetic and environmental factors [2]. Obesity and other lifestyle factors including diet are main contributors towards an increased risk of developing high blood pressure [3]. Adopting lifestyle changes, including a healthy diet, are integral to hypertension management. Dietary recommendations include a diet rich in plant-based foods, wholegrains, low-fat dairy products, and lowering sodium dietary intake within normal limits, for the prevention and management of hypertension [4]. This is promoted by the Dietary Approaches to Stop Hypertension (DASH) diet, described over 20 years ago [5,6] in subjects with hypertension. The effects of the DASH diet may be due to increased protein intake, increased dairy intake or reduction in sodium intake. Research across diverse populations suggests that well-formulated plant-based diets as sources of protein can help to manage blood pressure [7,8]. Different food groups can contribute to managing the risk of hypertension [9], with processed meat and wholegrains highlighted to support current dietary guidance for prevention. Plant based diets offer not only health benefits but also offer a public health strategy [10] to promote an environmentally sustainable diet. A shift towards less reliance on animal products, and more towards a plant-based diet can reduce the production of greenhouse gas emissions to address global warming. Other nutrients and dietary patterns have been explored to as dietary approaches to manage hypertension, including the Mediterranean diet. Systematic reviews [11,12] highlight a small but favourable effect of this diet pattern on hypertension. This diet is synonymous with olive oil consumption and the role of this on arterial blood pressure has been highlighted in the Greek EPIC cohort [13], which also highlighted the importance of cereal intake. There are some limited data on the role of wholegrains and dietary fibre on blood pressure, and specific dietary fibres. Cross-sectional studies and meta-analysis [13,14] suggest that an increase in dietary fibre lowers BP in subjects that are hypertensive, particularly for insoluble fibre, potentially attributed to the insulin lowering and effects on endothelial function.

While certain plant foods have been extensively researched, plant-based proteins have received comparatively less research attention. Legumes are a common source of plant-based protein; they are recognized as either the seed or fruit of a plant in the Fabaceae family, plants which produce a pod with a seed inside. Foods classified as legumes include peas, beans, lentils, peanuts, edamame, and common processed food products such as canned baked beans in tomato sauce [15,16]. Legumes as part of a plant-based diet, are encouraged as a protein-rich food to be increased in consumption to promote both human and planetary health [10]. Legumes contain various components linked to lower blood pressure, including dietary fibre, bioactive peptides, and flavonoid polyphenols [17,18,19,20,21].

However, limited evidence exists on the potential blood pressure lowering effect of legumes in humans. A 2013 systematic review and meta-analysis of dietary pulse intake and blood pressure from controlled feeding trials reported that greater legume intake had a significant inverse impact on blood pressure [22]. Further, a large 2020 prospective cohort study examined legume consumption and the risk of hypertension in Chinese adults. This study demonstrated a significant inverse association with hypertension with increasing legume intake [23]. However, legumes consumed in China are mostly part of traditional (often fermented) soya dishes and significantly differ to the type of legumes habitually consumed in the UK [24,25]. There are small randomized controlled trials in murine and human models where legume-based proteins demonstrated beneficial effects on blood pressure [26,27].

Against this background, we aimed to use prospective data collected in the 1990s to investigate the temporal associations between legume intake and blood pressure in a large cohort representative of the UK population, the European Prospective Investigation into Cancer-Norfolk (EPIC-Norfolk) cohort study.

## 2. Methods

### 2.1. Study Population

The study population was drawn from the participants of the Norfolk arm of the multi-centre European Prospective Investigation into Cancer (EPIC-Norfolk) project, which has been detailed elsewhere [28]. EPIC-Norfolk is a longitudinal cohort with extensive dietary information, starting from 1993. Participants were aged between 39 and 79 years at baseline, being recruited from participating General Practices in Norfolk and surrounding areas in the UK. The first health checks and 7-day food diaries were completed between 1993 and 1997, and second health checks took place between 1998 and 2000. Each health check was conducted at a clinic by trained nurses. More extensive details on the participants and their characteristics are available as previously reported on the EPIC-Norfolk website portal [29].

### 2.2. Exposure Variables

Total legume consumption was established by analysing the completed 7-day food diaries of the participants. These diaries were comprehensive 50-page booklets with consisted of an A5-sized booklet containing 17 sets of colour photographs representing portion sizes and instructions to guide the information to be reported. The diary had space for recording all food and drinks consumed over each 24-h period, running from midnight to the following midnight. The booklets had space for additional information, with full guidance provided to participants on describing and quantifying each item of food and drink. Information on using the diaries, and instructions on recalling food/drinks consumed over the previous day was provided by a trained nurse, who helped each participant fill in the first day of the 7-day diary. Completed diaries were comprehensively reviewed for accuracy and data were entered into specialist database software called DINER and DINERMO by trained data enterers [30,31]. This is further explained in the study website [32]. eDatabase specialists working on the EPIC-Norfolk dataset aggregated all legume consumption across individual foods and all legume-containing dishes. From this aggregation, total legume intake in average grams per day (as eaten, cooked weight) was calculated for each participant. For the initial analysis, legume intake was categorized according to specified percentiles of intake.

### 2.3. Outcome Measurement

The outcome measurement was hypertension at the second health check, undertaken between 1998 and 2000. Blood pressure was taken as a mean of two readings by trained nurses, both of which were taken after seated resting for five minutes. An Accutorr™ non-invasive oscillometric blood pressure monitor was used to take the blood pressure measurements. Hypertension was defined using the definition/guidelines of the National Institute for Health and Care Excellence (NICE): 140/90 mmHg and above (or systolic blood pressure over 140 mmHg or diastolic blood pressure over 90 mmHg) [32]. Both systolic and diastolic blood pressure cut-offs were used to identify hypertension, and either one being above the cut-off was categorized as having hypertension, as per NICE guidelines [32]. Any participant with hypertension (including those on medication) were included in this classification.

### 2.4. Confounding Variables

Data for a variety of covariates that could influence the results were identified and collected, with each covariate selected based upon the previous literature [33,34,35,36]. These covariates included age, sex, body mass index (BMI), and various dietary and lifestyle factors listed below. Habitual dietary intakes were considered for energy (kilojoules per day), protein (grams per day), sodium (milligrams per day), and alcohol (grams per day). Average intakes of vegetables, fruit, and red meat were also retrieved and analysed as markers of the participants’ dietary pattern. Lifestyle behaviours and baseline health status were considered, including smoking status, physical activity, high blood pressure at baseline, and anti-hypertensive medication use at the time of the second health check. Smoking status was categorized as current, former, or never. Physical activity (PA) was assessed by questionnaire (described in [37]), which combined habitual work and leisure activity to assign four categories as: inactive, moderately inactive, moderately active, and active. These classifications are described in Appendix A and Appendix B. Various socioeconomic factors were accounted for; education level was classified into no qualifications, ‘O’ level/GCSE, A-level, and degree or higher. Social class was stratified by occupational category, and classification categories included professional, managerial, skilled non-manual, skilled manual, semi-skilled, and non-skilled. Deprivation was considered as a variable by the Townsend deprivation index [38]. This is a composite index, to identify material deprivation, using a combination of four variables (the percentage of economically active residents over 16 years old who are unemployed; percentage of households with no car; percentage of households not owner occupied; percentage of households with more than one person per room—all at the enumeration district level). Other variables controlled for included anti-hypertensive medication use at the time of the second health check (yes/no) and self-reported high blood pressure at baseline (yes/no).

### 2.5. Statistical Analysis

Multiple logistic regression models were constructed to assess the impact of legume consumption on subsequent hypertension (yes/no). Logistic regression was selected over cox regression due to the temporal relationship between baseline legume intake and subsequent blood pressure. Since hypertension is not a discrete event, a cross-sectional analytical approach was taken. All analyses were undertaken with IBM SPSS Statistics for Windows, version 25.0. The analyses were conducted for all participants, and the results further stratified into men and women. The analyses were presented as odds ratios alongside the *p*-values and 95% confidence intervals. Statistical significance was deemed as *p* < 0.05. The normality of the data was checked, with visual inspection of the histogram of distribution for each variable.

These analyses were stratified by sex and controlled for age, BMI, and other covariates of interest listed above. Model A adjusted for age and BMI, model B further adjusted for energy intake, red meat, alcohol, sodium, protein, fruit, and vegetable intake. Model C adjusted for the same covariates as in model B, but also included smoking status, physical activity, high blood pressure at baseline, and anti-hypertensive medication use at the time of the second health check. Finally, model D adjusted for the above covariates (as in model C) with added socioeconomic variables: education level, social class, and deprivation index. The first analysis assessed legume intake by pre-specified percentile cut off points.

Informed by the results from the initial analysis and the intake levels that conferred the lowest odds of hypertension, a binary logistic regression analysis was undertaken by every 1-g increase in legume intake, commencing at 45 g/day. The purpose of this analysis was to elucidate the legume dietary intake threshold that confers benefit for lowering the subsequent odds of hypertension.

Further analysis was applied to identify the most effective daily range of legume intake for reduced hypertension odds. Various legume intake ranges, selected based on the results of the incremental binary logistic regression analyses, were assessed against the lowest range of legume intake (<10 g/day) using multivariate logistic regression model D.

## 3. Results

### 3.1. Participants

The participant flow for the current study is detailed in Figure 1. The characteristics for all participants are presented in Table 1, and is also summarised by sex, with *p*-values. The mean age (± SD) of participants at baseline was 58.0 ± 8.9 years (58.7 ± 8.9 and 57.4 ± 8.8 years for men and women respectively). Men were significantly older at baseline (*p* < 0.001) and had a 29.2% higher intake of legumes (*p* < 0.001). Men also had higher energy (+27.5%, *p* < 0.001), alcohol (+74.1%, *p* < 0.001), protein (+23.0%, *p* < 0.001), sodium (+27.5%, *p* < 0.001), and red meat (+36.0%, *p* < 0.001) intake. Additionally, men had slightly higher systolic (3.3%, *p* < 0.001) and diastolic (+4.7%, *p* < 0.001) blood pressure at baseline. In contrast, women had a higher intake of fruit and vegetables than men (+6.9%, *p* < 0.001). There were no significant differences in physical activity level or self-reported high blood pressure at baseline between sexes. No significant differences were seen in current smoking rates, though significantly more men were former smokers (+53.8%, *p* < 0.001). Furthermore, men tended to have a higher education level and social class categorized by occupation.

### 3.2. Legume Intake

Table 2 displays the impact of legume intake on subsequent hypertension odds with a median follow-up time of 3.7 years (range: 2.1–6.6 years) using pre-specified percentile cut off points for all participants and stratified by sex. Multivariable adjusted odds ratios and 95% confidence intervals are presented for different models. There was a significant temporal association between higher consumption of legumes and reduced subsequent odds of hypertension; intakes at the 97th percentile (>45–66 g/day) had the lowest odds and the 3rd and 5th percentile (<3.5 g/day) had the highest. In the unadjusted model, these temporal associations were statistically significant for all participants and this statistical significance held true for both sexes when sex-specific analyses were undertaken. In the fully adjusted model D, which adjusted for additional dietary, lifestyle, socioeconomic, and baseline health covariates, these results were only statistically significant in all participants and women. Although a similar trend was observed in men, it failed to reach statistical significance.

Table 3 presents the results of univariate logistic regression analyses by daily legume intake by each 1-g increase in intake, starting at 45 g per day. The analysis results displayed a trend where initially the odds ratios of hypertension decreased, and the confidence intervals narrowed with incremental increased intake. The association between legume intake and reduced odds of hypertension became statistically significant when passing the >54 g per day threshold in all participants and women. Again, while a similar trend existed for men, it did not reach statistical significance.

Table 4 indicates the results of multivariate logistic regression analyses, with all intake levels being compared to a reference category of the lowest consumers of legumes (<10 g/day). The results of this analysis demonstrated that legume intake in the range of 55 to 70 g/day was most significantly associated with lower odds of subsequent hypertension (OR: 0.57; 95% CI: 0.37, 0.88). When conducting sex-specific analysis, the odds of subsequent hypertension in the 55–70 g/day intake range were (OR: 0.64; 95% CI: 0.38, 1.04) and (OR: 0.32; 95% CI: 0.12, 0.88) in men and women, respectively.

## 4. Discussion

In this large UK prospective population-based study, we found a significant association between higher dietary legume intake and subsequent odds of hypertension over a median 3.7 years of follow up. The most significant effect range was observed with daily consumption between 55 and 70 g/day, and this intake range was associated with statistically significant lower odds of hypertension in all participants (43% reduced risk) and women (68% reduced risk). Although a similar trend was observed in men, it did not reach statistical significance. These data suggest a threshold effect, but this cannot be tested in the current cohort; this requires more detailed research in a larger cohort so that the amount and type of legume intake can be fully explored, preferably with higher intakes. Indeed, these data also highlight that low intakes of legumes are not associated with prevention of hypertension.

There are several plausible explanations for our findings, including the difference between male and female participant numbers in the cohort itself, since women (4030) outnumbered men (3492) by approximately 16%, thus providing more statistical power to detect significance. Furthermore, all else being equal, men and women have slightly different blood pressure at the same age. At baseline in this cohort, blood pressure was slightly higher in men than women. A study on gender difference in blood pressure regulation showed that blood pressure increased in both sexes with advancing age, but that men had higher 24-h mean blood pressure prior to age 70 [39]. Thus, a higher degree of blood-pressure-lowering would be required for men to fall under the hypertension threshold. This may partly explain the sex-specific differences in lower odds of hypertension by legume consumption.

Much of the published work on diet and health are established from epidemiological data, where a variety of bioactive compounds in plant foods are found to convey health benefits. Higher levels of legume consumption could be indicative of a more balanced or healthy diet and there may be interactive effects between eating legumes and a diet high in fruits and vegetables, wholegrains and low in saturated fats. There are multiple plausible mechanisms through which legume consumption could lower blood pressure. Legumes are rich in protein, fibre, B vitamins, minerals (including potassium and magnesium), and polyphenols. Numerous studies have shown that the bioactive peptides in legumes are beneficial for health (e.g., [40,41]) offering antioxidant, anti-cancer, and anti-inflammatory effects. Legumes offer hypotensive effects by inhibiting angiotensin-converting enzyme (ACE) activity, thus relaxing veins and arteries, and also increased nitric oxide production. Nitric oxide (NO) is a small free radical molecule. Legumes are a significant dietary source of L-arginine, which is a semi-essential amino acid, oxidized by the NO synthase enzymes to form NO which elicits multiple potentially beneficial effects linked to the cardiovascular system. Decreased ACE activity and nitric oxide can each both lower blood pressure and simultaneously activating these two pathways may have greater effect for health [40,41].

These data were collected more than 20 years ago and are used to show the correlation been diet and health parameters from a unique longitudinal cohort with high quality nutrient intake, validated and assessed from the food diary method. Most of the participants were meat eaters (97%), with very few vegetarian or vegan. However, habitual diets have changed considerably in the UK over this timeframe. More recent dietary intake data, from the UK National Diet and Nutrition Survey (NDNS) program, running 2008–2019, the mean (SD) legume intake within the United Kingdom was 26.7 ± 29.6 g/day [42]. This is slightly higher than that reported in the EPIC cohort, but still well below our suggested amount of above 50 g/d for prevention of hypertension. Consumers are more aware of the issues of environmental sustainability, water use, and air miles to influence food choice, but there are barriers to move towards a plant-based diet, including that is inconvenient, takes more time consuming to prepare and requires more cooking skills [43]. Legumes are not necessarily the first choice of protein source for people transitioning to a plant-based diet. Plant-based foods that mimic meat products, such as burgers and sausages, require fewer cooking skills [43]. Ironically, often these processed plant-based alternatives are high in sodium [44], which is not ideal for an anti-hypertensive diet. Legumes have also historically been associated with the presence of anti-nutrients (or bio-active non-nutrients), which, if processed inappropriately, can have unwanted health effects, such as toxicity or legume-related food allergies (e.g., peanuts or soyabean) [45].

Considering the wider literature there are some similarities between this study and the prospective cohort on legume consumption and hypertension risk reported in Chinese men and women, by Guo et al. [23]. In this cohort, quartile 3 of legume intake (median: 60.3 g/day) significantly lowered the risk of hypertension more than quartile 4 (median: 97 g/day). Notably, this finding appears to concur with the present study’s results, with the largest beneficial reduction in hypertension seen at an equivalent intake level. It is beyond the scope of this paper to investigate potential reasons why this range may be the most efficacious for lowering blood pressure; it is likely multifactorial, including the simple possibility that individuals consuming the highest intakes of legumes may be displacing intake from other beneficial food groups. This is an area for future research.

In this study, we did not analyse at the level of type of legumes consumed, as legume intake was aggregated into total amount. In this regard, it is highly likely that different types of legumes have varying effects due to their distinct nutritional characteristics; future research could help differentiate the effects of different legumes typically consumed within the UK.

There are several strengths of this study. This is the first large cohort to investigate the links between legume intake and blood pressure in a Western population, using the gold standard prospective 7-day diary dietary assessment method. EPIC-Norfolk presents an extensive collection of 7-day diary data, offering greater accuracy and reliability than recall methods such as food frequency questionnaires (FFQ). For instance, a study compared these two dietary assessment methods for accuracy against biological markers of intake, and the results demonstrated that the error variance and regression dilution were much higher for, suggesting lower reliability [46].

Furthermore, the study was able to control for a wide range of demographic, lifestyle, socioeconomic, and medications which can play a significant modifying effect on blood pressure. Since the study was a longitudinal prospective cohort study, the temporal relationship shown between legume consumption and hypertension also makes reverse causality unlikely. This is also the first UK study to support guidance to move towards the increased inclusion of legumes as part of a healthy and sustainable plant-based diet [10]. For example, the EAT-Lancet authors suggested 50 g of beans, lentils, and peas and 25 g of soya beans a day [17]. Overall, our data support the health benefit of increasing legume intake in the UK population, but that the population need support on how to achieve this dietary guidance. As mentioned, according to UK NDNS data, mean legume intake across the UK is 26.7 g/day. Thus, doubling this intake would bring it closer to the beneficial 55–70 g/day range; approximately ¾-1 cup of cooked legumes.

It is important to note the limitations of this study. Despite the size of the EPIC-Norfolk study population, the number of participants with legume consumption data who met the inclusion criteria was comparatively low (7522). Further, mean legume intake was only 17.3 g/day across all participants, so consumption was generally low across the cohort. It is unlikely that a larger collection of legume consumption data would change the observed trends and observations, but it may have allowed greater confidence in the findings through a larger sample size’s impact on statistical power, particularly for men. The median follow-up within our cohort was relatively short (3.7 years), and longer studies would be ideal to follow the relationship between diet and health.

To limit the possibility of residual confounding, extensive efforts were made to control for potential confounders. However, we were not able to account for residual confounding and the effects of unknown confounders. For example, intake of micronutrients magnesium and potassium are linked to blood pressure, and these were not accounted for. Although healthy responder bias can be a downside of cohort studies, the cohort is similar to other UK cohorts and is representative of the UK population. In addition, the internal relationship between legume consumption and blood pressure is unlikely to be affected by selection bias.

## 5. Conclusions

In conclusion, findings from the EPIC-Norfolk cohort suggest that further guidance to consume more legumes may have clinical value for lowering hypertension risk. Further research is needed to confirm these results and to better understand the mechanisms by which level of legumes consumption that can have impact on hypertension. In the meantime, alongside following conventional guidance on reducing hypertension risk, we recommend the greater uptake and promotion of legume consumption, ideally in the range identified as being most significantly associated with benefit identified in this work, which is approximately a cup serving per day.

## Figures and Tables

**Figure 1 nutrients-14-03363-f001:**
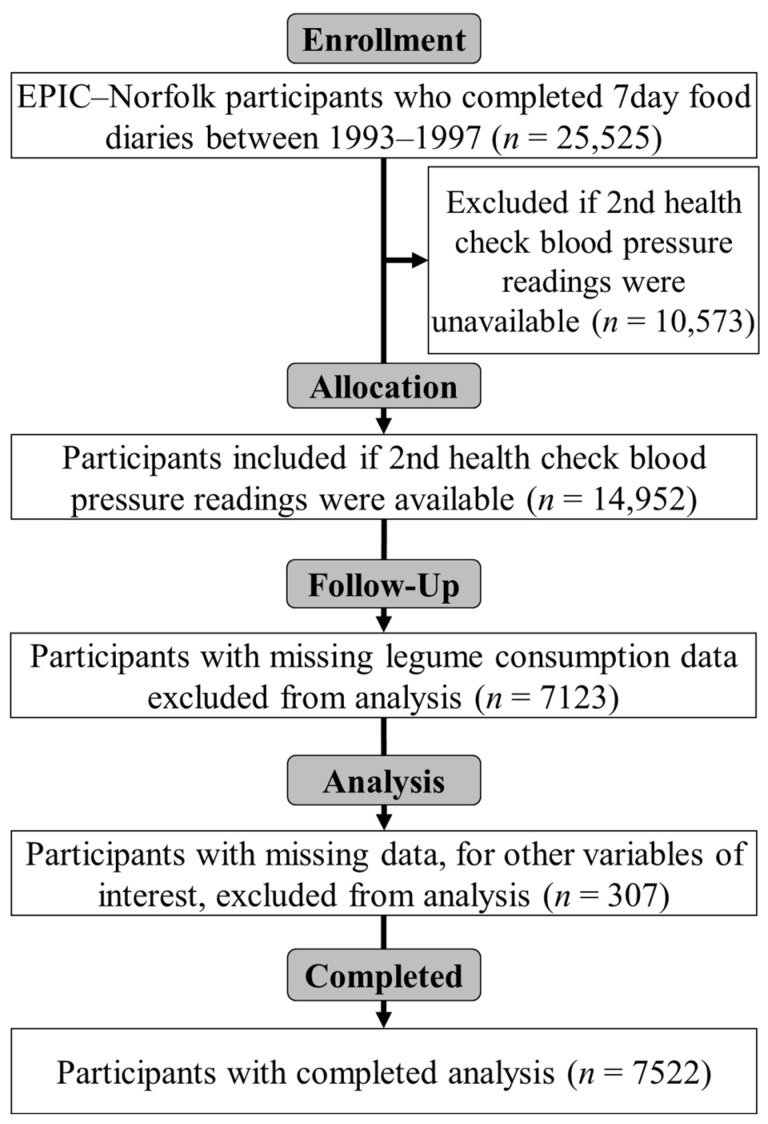
Consolidated Standards of Reporting Trials (CONSORT) diagram summarizing participant flow with the sizes (*n*) of initial and analysed groups.

**Table 1 nutrients-14-03363-t001:** Mean (± SD) overall and sex-specific sample characteristics of 7522 participants aged 39–79 years old at baseline (1993–1997) in the EPIC-Norfolk cohort.

	All (*n* = 7522)	Men (*n* = 3492)	Women (*n* = 4030)	*p*
Mean (SD)				
Age at entry (years)	58.0 (8.9)	58.7 (8.9)	57.4 (8.8)	<0.001
BMI (kg/m^2^)	26.0 (3.7)	26.4 (3.2)	25.7 (4.1)	<0.001
Legume intake (g/day)	17.3 (16.3)	20.0 (18.6)	14.9 (13.6)	<0.001
Energy intake (kJ/day)	8505 (2144)	9770 (2031)	7409 (1553)	<0.001
Alcohol intake (g/day)	12.3 (17.0)	17.1 (20.8)	8.1 (11.3)	<0.001
Protein intake (g/day)	75.3 (17.5)	84.7 (17.0)	67.2 (13.4)	<0.001
Sodium intake (mg/day)	2924 (844)	3359 (845)	2546 (636)	<0.001
Fruit and vegetable intake (g/day)	333 (162)	320 (158)	343 (164)	<0.001
Red meat intake (g/day)	33.2 (26.7)	39.7 (29.4)	27.6 (22.7)	<0.001
Systolic blood pressure (mmHg)	134.5 (17.8)	136.8 (16.9)	132.4 (18.3)	<0.001
Diastolic blood pressure (mmHg)	82.2 (11.1)	84.3 (10.9)	80.4 (1.0)	<0.001
Townsend Deprivation Index (index)	−2.15 (2.0)	−2.19 (2.1)	−2.12 (2.1)	0.144
Number (%)				
Self-reported high blood pressure	942 (12.5)	457 (13.1)	485 (12.0)	0.173
Smoking status at baseline				<0.001
Current smoker	697 (9.3)	338 (9.7)	359 (8.9)	
Former smoked	3166 (42.1)	1902 (54.5)	1264 (31.4)	
Never smoked	3659 (48.6)	1252 (35.8)	2407 (59.7)	
Physical Activity Level at baseline:				<0.001
Inactive	1933 (25.7)	923 (26.4)	1010 (25.1)	
Moderately inactive	2199 (29.2)	884 (25.3)	1315 (32.6)	
Moderately active	1832 (24.4)	857 (24.5)	975 (24.2)	
Active	1558 (20.7)	828 (23.7)	730 (18.1)	
Education level:				<0.001
No education	2492 (33.1)	954 (27.3)	1538 (38.1)	
GCSE/O Level	823 (10.9)	308 (8.8)	515 (12.8)	
A-Level	3176 (42.3)	1670 (47.8)	1506 (37.4)	
Degree or higher	1031 (13.7)	560 (16.0)	471 (11.7)	
Social class by occupation:				<0.001
Unknown	52 (0.7)	25 (0.7)	27 (0.7)	
Professional	349 (4.6)	277 (7.9)	72 (1.8)	
Manager	2504 (33.3)	1365 (39.1)	1139 (28.3)	
Skilled non-manual	2058 (27.4)	432 (12.4)	1626 (40.3)	
Skilled manual	1109 (14.7)	856 (24.5)	253 (6.2)	
Semi-skilled	1127 (14.9)	459 (13.1)	668 (16.6)	
Non-skilled	323 (4.3)	78 (2.2)	245 (6.1)	

A-level, Advanced level qualification; EPIC, European Prospective Investigation into Cancer and Nutrition cohort; GCSE, General Certificate of Secondary Education; mmHg, millimetres of mercury.

**Table 2 nutrients-14-03363-t002:** Odds of hypertension (defined as either systolic or diastolic blood pressure over 140/90) by daily legume intake percentile: multivariate odds ratios (OR) and 95% confidence intervals (95% CI).

Percentile (g/day)	Unadjusted Model		Model A		Model B		Model C		Model D	
	OR	95% CI	OR	95% CI	OR	95% CI	OR	95% CI	OR	95% CI
3rd (1.84)	1.47 **	1.12, 1.91	1.34 *	1.02, 1.77	1.33 *	1.01, 1.76	1.34 *	1.01, 1.77	1.36 *	1.03, 1.80
5th (3.15)	1.43 ***	1.16, 1.77	1.33 *	1.07, 1.65	1.32 *	1.06, 1.65	1.34 **	1.08, 1.68	1.37 **	1.10, 1.71
10th (4.14)	1.11	0.96, 1.29	1.02	0.88, 1.20	1.04	0.88, 1.21	1.04	0.89, 1.22	1.05	0.90, 1.24
50th (13.8)	0.96	0.87, 1.05	1.02	0.92, 1.12	1.00	0.91, 1.11	1.01	0.91, 1.12	1.00	0.90, 1.10
90th (33.9)	0.84 *	0.72, 0.99	0.956	0.81, 1.12	0.95	0.80, 1.12	0.95	0.80, 1.12	0.93	0.79, 1.10
95th (45.2)	0.83	0.67, 1.04	0.94	0.75, 1.18	0.92	0.73, 1.16	0.94	0.75, 1.19	0.93	0.74, 1.17
97th (56.6)	0.65 **	0.49, 0.87	0.72 *	0.54, 0.97	0.69 *	0.52, 0.94	0.72 *	0.53, 0.97	0.71 *	0.52, 0.96
Men										
3rd (2.55)	1.66 *	1.12, 2.47	1.53 *	1.01, 2.30	1.43	0.94, 2.16	1.38	0.91, 2.09	1.42	0.94, 2.17
5th (3.49)	1.26	0.93, 1.70	1.15	0.84, 1.57	1.08	0.79, 1.49	1.06	0.77, 1.46	1.09	0.79, 1.51
10th (5.23)	1.16	0.93, 1.45	1.07	0.85, 1.35	1.04	0.82, 1.31	1.03	0.81, 1.30	1.06	0.83, 1.34
50th (14.9)	0.92	0.80, 1.05	0.96	0.83, 1.10	0.97	0.84, 1.12	0.98	0.85, 1.13	0.96	0.84, 1.11
90th (39.5)	0.79*	0.63, 0.99	0.89	0.71, 1.12	0.91	0.72, 1.16	0.93	0.74, 1.19	0.91	0.72, 1.16
95th (54.6)	0.69*	0.50, 0.94	0.76	0.55, 1.06	0.78	0.56, 1.09	0.82	0.58, 1.14	0.80	0.57, 1.12
97th (66.2)	0.81	0.54, 1.21	0.90	0.60, 1.35	0.94	0.62, 1.43	1.00	0.66, 1.53	0.98	0.65, 1.50
Women										
3rd (1.84)	1.52 *	1.09, 2.12	1.31	0.92, 1.87	1.31	0.92, 1.86	1.35	0.95, 1.93	1.36	0.95, 1.95
5th (2.75)	1.62 ***	1.23, 2.15	1.51 **	1.13, 2.03	1.50 **	1.12, 2.02	1.55 **	1.15, 2.10	1.58 **	1.17, 2.13
10th (3.67)	1.09	0.90, 1.34	1.02	0.83, 1.27	1.02	0.82, 1.26	1.04	0.84, 1.29	1.05	0.84, 1.30
50th (11.4)	0.88 *	0.77, 0.99	0.97	0.85, 1.12	0.98	0.85, 1.13	0.99	0.86, 1.14	0.99	0.86, 1.14
90th (30.2)	0.83	0.66, 1.03	0.93	0.74, 1.18	0.95	0.75, 1.20	0.97	0.76, 1.23	0.97	0.77, 1.24
95th (38.3)	0.61 **	0.44, 0.85	0.73	0.52, 1.03	0.74	0.53, 1.05	0.75	0.53, 1.06	0.75	0.53, 1.07
97th (44.8)	0.58 *	0.38, 0.88	0.70	0.45, 1.08	0.70	0.46, 1.10	0.73	0.46, 1.14	0.73	0.47, 1.14

Statistical significance: * = *p* < 0.05, ** = *p* < 0.01, *** = *p* < 0.001. Multivariate model A: adjusted for age and BMI; multivariate model B: model A + adjusted for energy intake, red meat intake, alcohol intake, sodium intake, fruit and vegetable intake, and total protein intake; multivariate model C: model B + adjusted for smoking status, physical activity, self-reported blood pressure at baseline, and anti-hypertensive medication use at time of the second health check; multivariate model D: model C + adjusted for education level, social class, and deprivation index.

**Table 3 nutrients-14-03363-t003:** Odds of hypertension by grams-per-day legume intake, starting above 45 g per day, using fully adjusted multivariate model D: odds ratios (OR) and 95% confidence intervals (95% CI).

Legume Intake (g/day)	All (*n* = 7522)		Men (*n* = 3492)		Women (*n* = 4030)	
	OR	95% CI	OR	95% CI	OR	95% CI
>46	0.95	0.75, 1.21	0.99	0.75, 1.32	0.76	0.48, 1.20
>47	0.96	0.75, 1.23	0.99	0.74, 1.33	0.79	0.50, 1.25
>48	0.92	0.72, 1.19	0.97	0.72, 1.30	0.73	0.45, 1.18
>49	0.90	0.70, 1.16	0.96	0.71, 1.30	0.63	0.37, 1.07
>50	0.90	0.69, 1.17	0.96	0.70, 1.31	0.63	0.37, 1.09
>51	0.89	0.68, 1.16	0.93	0.68, 1.28	0.63	0.36, 1.10
>52	0.84	0.64, 1.11	0.88	0.64, 1.22	0.61	0.35, 1.09
>53	0.80	0.61, 1.07	0.86	0.62, 1.19	0.55	0.30, 1.02
>54	0.74 *	0.55, 0.99	0.83	0.60, 1.16	0.37 **	0.18, 0.75
>55	0.72 *	0.54, 0.97	0.79	0.56, 1.11	0.40 *	0.20, 0.82
>56	0.70 *	0.52, 0.94	0.76	0.54, 1.07	0.40 *	0.20, 0.82
>57	0.73 *	0.54, 0.99	0.79	0.56, 1.12	0.43 *	0.21, 0.87
>58	0.79	0.58, 1.09	0.84	0.59, 1.21	0.48 *	0.24, 1.00
>59	0.79	0.57, 1.08	0.86	0.60, 1.24	0.44 *	0.21, 0.94
>60	0.77	0.56, 1.08	0.83	0.57, 1.21	0.47	0.22, 1.02
>61	0.82	0.58, 1.15	0.88	0.60, 1.30	0.50	0.24, 1.08
>62	0.82	0.58, 1.16	0.91	0.62, 1.34	0.45 *	0.20, 0.99
>63	0.86	0.60, 1.22	0.98	0.66, 1.47	0.41 *	0.18, 0.95
>64	0.87	0.61, 1.23	0.97	0.65, 1.46	0.44	0.19, 1.02
>65	0.88	0.62, 1,27	1.01	0.67, 1.53	0.41	0.17, 1.01
>66	0.89	0.61, 1.28	1.01	0.67, 1.54	0.42	0.17, 1.03

Statistical significance: * = *p* < 0.05, ** = *p* < 0.01.

**Table 4 nutrients-14-03363-t004:** Odds of subsequent hypertension compared to the lowest legume consumers (<10 g/day) by daily legume intake range using fully adjusted multivariate model D: odds ratios (OR) and 95% confidence intervals (95% CI).

Legume Intake (g/day)	All (*n* = 7522)		Men (*n* = 3492)		Women (*n* = 4030)	
	OR	95% CI	OR	95% CI	OR	95% CI
10–24.9	0.90	0.89, 1.11	0.98	0.83, 1.16	0.97	0.84, 1.13
25–39.9	1.08	0.92, 1.27	0.94	0.75, 1.17	1.23	0.96, 1.56
40–54.9	1.01	0.78, 1.32	1.00	0.71, 1.41	0.92	0.60, 1.42
55–69.9	0.57 *	0.37, 0.88	0.64	0.38, 1.03	0.32 *	0.12, 0.88
70–84.9	0.70	0.34, 1.46	0.58	0.26, 1.31	0.87	0.17, 4.53
85–99.9	1.29	0.64, 2.60	1.32	0.59, 2.93	0.95	0.18, 4.94
>100	0.89	0.44, 1.78	1.12	0.51, 2.47	0.18	0.02, 1.50

Statistical significance: * = *p* < 0.05.

## Data Availability

Data described in the manuscript, code book, and analytic code will be made available upon request.

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
