# Peer review of "Association between Legume Consumption and Risk of Hypertension in the European Prospective Investigation into Cancer and Nutrition (EPIC)-Norfolk Cohort"

_nutrients, 2022, doi:10.3390/nu14163363_

Round 1
Reviewer 1 Report
Overall a well written paper that was enjoyable to read.
The background provides incidence of CVD where the paper is not looking at CVD but rather just blood pressure (BP). I would suggest it be written with about BP not CVD though CVD can be referenced. Minor points but Indeed and regrettably can be removed lines 42 and 43.
The background could also consider other foods / food groups with evidence based outcomes relating to BP e.g Mediterranean dietary patterns, soluble fibre, dietary, fatty acids. Vegetable intake is one factor taken into consideration in the analysis however.
The results are up to 24 years old. If they remain significant today, this needs to be discussed. Many changes in diet have occurred in this period. Growing awareness of sustainability, water use and plant based protein are all important to this discussion. Newer research on anti-nutrients in some legumes is also important to be considered.
Writing the paper as a more current review and citing the data collected from 1993-1997 as a baseline is more appropriate that writing as current research.
Analysis - if consumption is low, that may explain the lack of significance found. The finding is in fact not that increasing legume content was associated with lower BP. There was no significant difference at the highest and lowest extremes.
None-the-less, the health benefits of legumes and their role in the human diet and food chain is an increasingly, important area for research.
Author Response
Thank you for these helpful comments, we have amended the manuscript to account for all of these points.

Reviewer 2 Report
Please see attached word document.

Author Response
Thank you for the supportive comments about this manuscript. We have addressed all these points in the revised manuscript.

Reviewer 3 Report
The manuscript of Heartley and colleagues deals with investigating the relationship between legume intake and blood pressure in a cross-sectional analysis using the data from the Norfolk EPIC cohort. The manuscript is well written and discussed and the topic is interesting and surprisingly barely approached in the European population. The manuscript publication will help to promote legume intake which is extremely low in the European population. The manuscript has also several strengths that are a large number of subjects and the dietary assessment method, i.e., the 7-day diary. However, the median follow-up time is quite short (3.7 years) which is, in my opinion, weird since the cohort is still alive. In the title, the fact that the EPIC cohort is limited to that of Norfolk one should be mentioned. The authors didn’t take into account potassium and magnesium intake which are micronutrients related to blood pressure. Moreover, some parts of the manuscript deserve to be improved and clarified. The recording of dietary and physical data is not clearly explained. I understood that the health check took place twice. Was the same for the dietary data recording? Regarding the confounding variables, some information needs to be added: i) how was the physical activity measured? ii) how was the population categorized as inactive, moderately active and so on? iii) what is deprivation and how was calculated? please also add a reference. iv) did you check the normality of the data? it is unusual to present them in mean ± SD. In addition, how the legume intake data were processed: for instance, the legume intake can be reported as dry legumes or as cooked ones. How was the conversion handled? Moreover, were the meat replacers included in the legume intake calculation? Probably at that time they were not as popular as now but maybe someone consumed them. These products are produced mainly using legume proteins. Moreover, no information was added regarding if vegetarians were presented in the cohort. In the Discussion section, the part related to the mechanisms explaining the positive effect of legume intake should be improved.
Minor comments
Lines 88 and 95: is this the correct reference?
Table 1: please add how the data were presented.
Lines 211-216: is this text the footnotes of table 2?
line 229: why did you choose 10 g/day?
Author Response
Thank you for your comments and we have addressed these in the revised manuscript.

Round 2
Reviewer 1 Report
The paper is greatly improved by the changes made. Thank you to the authors for addressing all area of concern.